# Thermodynamic and structural investigation of oily wastewater treatment using peach kernel and walnut shell based activated carbon

Atef El Jery[1]*, Khaled Mohamed Khedher[2,3], Hayder Mahmood Salman[4], Nadhir Al-Ansari[5]*, Saad Sh. Sammen[6], Miklas Scholz[7,8,9,10,11,12]

1 Department of Chemical Engineering, College of Engineering, King Khalid University, Abha, Saudi Arabia, 2 Department of Civil Engineering, College of Engineering, King Khalid University, Abha, Saudi Arabia, 3 Department of Civil Engineering, High Institute of Technological Studies, Mrezgua University Campus, Nabeul, Tunisia, 4 Department of Computer Science, Al-Turath University College, Al Mansour, Baghdad, Iraq, 5 Civil, Environmental and Natural Resources Engineering, Lulea University of Technology, Lulea, Sweden, 6 Department of Civil Engineering, College of Engineering, University of Diyala, Baqubah, Diyala Governorate, Iraq, 7 Atene KOM, Berlin, Germany, 8 School of Science, Engineering and Environment, Newton Building, The University of Salford, Salford, Greater Manchester, United Kingdom, 9 Department of Civil Engineering Science, School of Civil Engineering and the Built Environment, Kingsway Campus, Aukland Park, University of Johannesburg, Johannesburg, South Africa, 10 Department of Town Planning, Engineering Networks and Systems, South Ural State University, Chelyabinsk, Russia, 11 Nexus by Sweden, Västerås, Sweden, 12 Kunststoff-Technik Adams, Specialist Company According to Water Law, Elsfleth, Germany

* nadhir.alansari@ltu.se (NAA); hyderalturath@gmail.com (AEJ)

**Data Availability Statement:** To ensure transparency and facilitate further research, all data

## Abstract

Despite the many articles about activated carbon with different precursors in adsorption process, no in-depth research has been carried out to understand the causes of the difference in surface adsorption characteristics of activated carbon with different precursors and different activation processes. In this work, the ability of two active carbon adsorbents made of walnut shell and peach kernel by two chemical and physical methods (totally 4 different types of activated carbon) in treatment of oily wastewater including diesel, gasoline, used oil or engine lubricant has been compared. The results show that the chemical activated peach carbon active with 97% hardness has provided the highest hardness and physical activated walnut carbon active has obtained the lowest hardness value (87%). It is also found that peach activated carbon has a higher iodine number than walnut activated carbon, and this amount can be increased using chemical methods; Therefore, the highest amount of Iodine Number is related to Peach activated carbon that is made by chemical method (1230 mg/g), and the lowest amount of iodine number is seen in walnut activated carbon that is made by physical method (1020 mg/g). moreover, the pore diameter of physical activated carbon is lower than chemical activated carbon in all cases. So that the pore diameter of chemical activated peach carbon active is equal to 22.08 μm and the measured pore diameter of physical activated peach carbon active is equal to 20.42 μm. These values for walnut are obtained as 22.74 μm and 21.86 μm, respectively. Furthermore, the temperature and pH effects on the adsorption of different synthesized oily wastewater was studied and it was found that a

necessary to replicate the findings of this study are presented within the paper. Additionally, the raw data is readily available from the corresponding author upon reasonable request. We are committed to providing the resources necessary for thorough evaluation and potential replication of our work.

**Funding:** This work was funded by the King Khalid University, Abha, Saudi Arabia. The appreciation to the Deanship of Scientific Research at King Khalid University for funding this work through Large Groups Project under grant number (R.G.P. 2/57/44). The funders had no role in study design, data collection and analysis, decision to publish, or preparation of the manuscript.

**Competing interests:** The authors have declared that no competing interests exist.

decrease in adsorption can be seen with an increase in temperature or decreasing the pH value, which can be referred to this fact that the process of adsorption is an exothermic process. Finally, to analyze the compatibility of adsorption isotherms with experimental data and to predict the adsorption process, three different isotherms named Langmuir, Temkin, and Freundlich isotherms were applied and their parameters were correlated. The correlation results show that the Langmuir isotherm had the best correlation in all cases compared to the Freundlich and Temkin isotherms, based on the correlation coefficient, and the calculated $R^2$ values which was greater than 0.99 in all the studied cases.

# 1. Introduction

The sea is one of the most important natural habitats, which is also a valuable source of food supply. Therefore, in addition to causing irreparable damage to the environment, contamination of this habitat directly or indirectly affects human health. Today, the production of oil and gas in the sea as well as the transportation of all kinds of fossil fuels has caused oil pollution in different areas of the sea to be more and more problematic [1].

In other words, spilling oil into the sea has become one of the biggest global concerns that has caused environmental problems [2]. It is worth noting that solving this environmental problem has also caused many economic problems. So far, many methods have been used to remove this type of pollution, including biological (including biological cleaning) [3], chemical (such as burning in place and using gelling agents) [4] and physical (including adsorbents, and skimmers) [5].

Choosing the best method to deal with depends on many factors, such as: polluting substance type, water conditions, and location of contamination; it should be emphasized that each method has its own advantages and disadvantages that influence the choice of each method [6].

Among the methods that have been applied for treatment of oil pollutants from water, the adsorption process has attracted a lot of attention. Today, the use of adsorbent materials is one of the most cost-effective and economical methods of removing oil pollution from the ground or polluted waters. In other words, these materials have the ability to transfer oil from the liquid phase to the solid phase, thus enabling the removal of oil from the sea [7, 8]. Among the effective factors in choosing adsorbents for the removal of petroleum compounds, availability, affordability, hydrophobicity, biodegradability, adsorption rate and adsorption capacity are the most important factors [9]. Adsorbents are generally divided into three categories: natural organic adsorbents, mineral adsorbents and synthetic adsorbents. natural organic adsorbent materials are relatively cheap and abundantly available and have a very high adsorption capacity [10]. Aerogel, perlite and mineral soils can be mentioned among the most important mineral adsorbents [11]. On the other hand, polypropylene and polyurethane are the most common synthetic organic materials in removing oil pollution due to their hydrophobicity [12].

Activated carbon refers to a group of carbon-based materials with porous structure, high adsorption capacity, and surface reactivation capability which can be commercially produced using many carbon-rich organic materials, such as wood [13] and coal [14]. Due to the expensiveness of the precursor materials and the non-renewability of these materials, the production of carbon active using these kinds of materials, is also expensive and its use is limited on a large industrial scale [15]. Therefore, in recent years, a lot of researches have been carried out for the cost-effective development of activated carbon production [16–18]. Various types of industries and agricultural waste such as peaches, walnut shells, almond shells, hazelnut shells,

coconut shells, olive kernels, coffee shells, plum and apricot kernels, cherry kernels and grape seeds, sugarcane pulp, bamboo husks, rice husks, pomegranate seeds, sludge and industrial food sludge have been used as activated carbon precursors [19]. In addition to agricultural waste, polymeric and mineral materials such as worn tires [20] and waste newspapers [21] have also been applied to produce carbon active with specific purposes and limited use.

In general, two methods can be used to activate the carbon-based materials; physical activation process (PAP) and chemical activation process (CAP). The PAP is carried out in two stages using gases such as carbon dioxide, water vapor, nitrogen, or a mixture of these [22]. In this way, after the pyrolysis of the raw materials, a reaction takes place between the oxidizing gas and the inactivated carbon at a temperature above 977°C, which causes the pores to be activated. However, in the CAP, a chemical activating agent, such as $KIO_4$, $ZnCl_2$, $HNO_3$, $H_2O_2$, $K_2CO_3$, KOH, and $H_3PO_4$, in one step, at a much lower temperature, can activate the pores. The chemical method uses more expensive materials and on a smaller scale than the physical method [23].

Despite the many articles about activated carbon with different precursors, no in-depth research has been carried out to understand the causes of the difference in surface adsorption characteristics of the obtained carbon active with different precursors and different activation processes.

In this research, the activated carbon was obtained by PAP and CAP method with precursors of hard walnut shell and peach kernel. In the following, the performance of each of the obtained activated carbons in adsorption of oil from wastewater has been investigated using synthesized oily wastewater include diesel, gasoline, used oil or engine lubricant. FTIR and XRD analyzes and SEM images have been used to identify the functional groups of the synthesized activated carbon surface, and to measure the pore structure of the synthesized carbon active. Finally, the equilibrium adsorption of each of the obtained activated carbons in the adsorption of each of the petroleum effluents has been studied and compared using three adsorption isotherm models and discussed carefully to understand the adsorption mechanism.

## 2. Materials and methods

### 2.1. Materials and chemicals

The applied raw materials in this research are peach kernel and walnut hard shell (stony endocarp), which were used to produce the desired activated carbon. All the properties of activated carbon have been measured according to ASTM standard methods in a quality control laboratory. The oily wastewater was synthesized using diesel, gasoline, used oil or engine lubricant, with density of 0.85, 0.73, 0.93, and 0.89 kg/l, respectively. Tween80 was used as an emulsifier and in all experiments double distilled water was used.

### 2.2. Experimental

**2.2.1. Carbonization.**   Carbonization is the most important steps in the process of carbon active preparation, because in this step the initial porous structure is formed [24]. In this way, the granules are first dried at a temperature of 200°C to remove the excess amount of the binder agent and to obtain the required mechanical strength. The dried granules are carbonized at high temperatures up to about 800°C [25].

In the process of carbonization, non-carbon substances Including oxygen, hydrogen, and volatile substances are released, and the free carbons are arranged in groups and form graphite crystals. Due to the presence of pores between the crystals, the structure of the crystals is irregular. This process takes place at a temperature of 800°C in the furnace [26]. After the initial combustion, the walnut shell and peach kernel act as fuel and provide the necessary heat to

continue the pyrolysis process. At the temperature of about 500˚C, the pores in the structure of the synthesized carbon are obtained. However, it should be noted that almost all of the initially obtained pores are blocked by the bituminous material. To open the initially obtained pores, in order to use them as adsorbents, the activation step must be carried out subsequently.

**2.2.2. Activation.** The carbonized material can be activated using either a physical activation method (PAP) or chemical activation one (CAP). In the PAP method, which is a direct method, first, the raw material is carbonized at 600˚C in the presence of nitrogen, argon, etc. as an inert gas; then the carbon-based substances is activated applying carbon dioxide gas at high temperatures of 750–900˚C, water vapor at high temperatures of 900˚C, or air at temperatures of 300–400˚C [27]. In the activation operation of carbon-based substances at a temperature of 800 to 1000˚C in the presence of carbon dioxide, they participate in reactions in which carbon is consumed during two separate reactions; during the first reaction, carbon dioxide, water vapor, carbon monoxide and hydrogen are produced and during another reaction, carbon monoxide, which is a poisonous and dangerous substance, is consumed with water vapor and produces carbon dioxide and hydrogen [28]. These reactions, by removing the bituminous materials resulting from pyrolysis, cause the opening, expansion and internal connection of the carbon pores, and as a result, the internal surface of the adsorber increases [22].

In the chemical process, the raw material is mixed with a chemical substance and then this mixture is carbonized and activated in a furnace. During the carbonization, due to the possibility of tar formation, which fills the space between the crystals, the process can be limited. One of the most important reactions of the chemical decomposition of the raw material is the dehydration reaction, which causes the decomposition of the raw material by heat [23].

Various chemical activating agents have been applied to produce activated carbon. Through the activated carbon production process, zinc chloride is used in a rotary furnace at a temperature of 600–700˚C. The mixture is then activated with water and hydrochloric acid to recover zinc chloride, and then the product is dried and granulated [29].

To produce activated carbon using a chemical process with phosphoric acid as a chemical activator, carbon material can be mixed with phosphoric acid, and then this mixture is put into a rotary kiln, after the carbonization, the activated carbon can be reached at ambient temperature [30].

To produce activated carbon, using a chemical activation method, first peach kernels or walnut shells were crushed. Then a certain amount of zinc chloride ($ZnCl_2$) was added to the crushed raw material. The prepared solution was stirred for 6 hours at 110˚C. After rinsing, the resulting material was dried in air at ambient temperature for 48 hours. Then it was placed in at a temperature of 240˚C for 6 hours using a furnace and then placed under argon gas at a temperature of 600˚C for 60 minutes. Finally, the resulting mixture was washed with deionized water and chloride acid for activation. The resulting activated carbon was completely dried at 95˚C for further use [31].

**2.2.3. Adsorption test.** The surface adsorption will be happened when a material from one phase is selectively trapped on the surface of another material (liquid or solid). The adsorption processes are exothermic, so low temperature and high pressure are thermodynamically favorable. The surface adsorption is a reversible process and there are van der Waals forces between the adsorbing and adsorbent molecules with the adsorption energy of less than 50 kJ/mol. The van der Waals force is weak and there is no need for a chemical reaction to occur. So, in this type of adsorption, there is no chemical reaction and the adsorbed molecules are only collected on the solid surface. In the process of physical adsorption, due to existence of the van der Waals forces, there is no type of electron transfer. The adsorption process depends on the physical properties of the adsorbing and adsorbent molecules [32, 33]. In order to perform the adsorption test, a certain amount of adsorbent material was placed in a

solution with a certain pollution. The amount of adsorption was measured at different times and under different operating conditions using a scale with an accuracy of ±0.0001 g [34].

## 2.3. Isotherm models

In order to relationship investigation between the concentration of the adsorbed substance and the concentration of the solution at equilibrium condition, different adsorption isotherm models are used. The isotherm models can be used to understand the adsorption mechanism and to optimize the required adsorbent amount in a process [35].

Equilibrium surface adsorption isotherms, which show the adsorption process between two phases, are of particular importance. When the adsorbent is placed in contact with a pollutant solution, the pollutant concentration on the adsorbent surface increases until reaching thermodynamic equilibrium and then stabilizes at an equilibrium state. The obtained equilibrium value, known as the adsorption isotherm, is the primary basis for the design of adsorption systems [36].

In other words, these data can be used to compare different adsorbents, as well as to design and optimize chemical processes. To investigate the behavior of adsorbents in removing pollutants and drawing adsorption isotherms, experiments in discontinuous mode should be used and then should be investigated using different adsorption models, such as Langmuir, Freundlich and Temkin isotherms [37].

The Langmuir model, first introduced in 1918, is the first kinetically oriented theory presented for surface adsorption on a flat surface. In this theory, a continuous process of the molecules movement on the surface and also the removal of molecules occurs so that the rate of accumulation of surface molecules remains equal to zero at the equilibrium state [38].

One of the Langmuir isotherm assumptions is that the surface adsorption energy is constant and the same in all sites. In other words, the adsorbent structure is assumed to be homogeneous in the Langmuir theory and each adsorption site can adsorb only one molecule [39, 40].

The linear equation of Langmuir isotherm model is presented in Eq 1, where $q_m$ is the maximum adsorption capacity, and $C_e$ is the equilibrium concentration; so, a linear plot of $\frac{C_e}{q_e}$ versus $C_e$ can be used to confirm the validity of the Langmuir correlation.

$$\frac{C_e}{q_e} = \frac{C_e}{q_m} + \frac{1}{K_l q_m} \tag{1}$$

In Eq 1, the $q_e$ is the equilibrium adsorbent concentration. It should be emphasized that the Langmuir constant ($K_l$), is related to the strength of molecule adsorption on the adsorbent surface. Therefore, the larger $K_l$ is, the more surface is covered with adsorbed molecules.

The Freundlich isotherm [41] is another equation which can be used to describe the equilibrium data, which is presented as follows:

$$\ln q_e = \frac{\ln C_e}{n} - \ln K_F \tag{2}$$

Where $K_F$ and $n$ are temperature-dependent Freundlich isotherm constants. As it can be understood from Eq 2, plotting $\ln q_e$ in terms of $\ln C_e$ and calculating the slope and the intercept, the constants of the equation can be presented.

It is obvious that if $n = 1$, the Freundlich equation will be linear, and the larger the value of $n$, the more the isotherm deviates from the linear state and will have a non-linear behavior. The $K_F$ represents the irreversibility of the process. In this model, it is assumed that the surface is non-uniform in terms of energy distribution, and sites with the same energy, are placed next to each other. It is also assumed that each molecule is adsorbed on just one site [42].

The linearized Temkin isotherm is shown in Eq 3, where $R$ is the universal gas constant, $K_T$ and $b$ are adjustable parameters, and $T$ is the temperature. As it can be seen, plotting $q_e$ in terms of $\ln C_e$ and calculating the slope and the intercept, the constants of the equation can be presented [42].

$$q_e = \frac{RT}{b}\left(\ln C_e + \ln K_T\right) \tag{3}$$

## 3. Results and discussion

The characteristics of fabricated activated carbons from precursors of peach kernel and walnut shell using physical or chemical method are listed in Table 1. As can be seen in Table 1, Peach Carbon Active has a higher Iodine Number than Walnut Carbon Active, and this amount can be increased using chemical methods; Therefore, the highest amount of Iodine Number is related to Peach activated carbon which is made by chemical method, and the lowest amount of Iodine Number is seen in Walnut activated carbon which is made by physical method.

This behavior can also be seen in Hardness and Specific Surface Area, so that the chemical activated peach carbon active with 97% hardness has provided the highest hardness and physical activated walnut carbon active has obtained the lowest hardness value (87%). Also, the Specific Surface for chemical activated peach carbon active and physical activated peach carbon active are 1610 m²/g and 1150 m²/g, respectively. These values for walnut activated carbon are obtained as 1550 m²/g and 1110 m²/g, respectively.

However different behavior was observed in density, ash and pH value of the obtained carbon actives. It was resulted that the density of physical activated carbon is higher than chemical activated carbon in all cases. So that the density of chemical activated peach carbon active is equal to 0.3 kg/m³ and the density of physical activated peach carbon active is equal to 0.5 kg/m³. These values for walnut are obtained as 0.25 kg/m³ and 0.4 kg/m³, respectively. Therefore, the highest value of obtained density is related to physical activated peach carbon active and the lowest density value is related to chemical activated walnut carbon active.

The ash percentage of physical activated carbon is more than chemical activated carbon in all cases. So that the ash percentage of chemical activated peach carbon active is equal to 1% and the obtained ash of the physical activated peach carbon active is equal to 3.5%. These values for walnut are equal to 5% and 7%, respectively. Therefore, the highest amount of ash

**Table 1. Characteristics of fabricated activated carbons from precursors of peach kernel and walnut shell.**

| Parameter | Peach Activated Carbon | | Walnut Activated Carbon | |
|---|---|---|---|---|
| | Physical Activation | Chemical Activation | Physical Activation | Chemical Activation |
| **Iodine Number (mg/g)** | 1050 (mg/g) | 1230 (mg/g) | 1020 (mg/g) | 1110 (mg/g) |
| **Hardness** | 91% | 97% | 87% | 92% |
| **Density (kg/m³)** | 0.5 (kg/m³) | 0.3 (kg/m³) | 0.4 (kg/m³) | 0.25 (kg/m³) |
| **Ash (%)** | 3.5% | 1% | 7% | 5% |
| **Humidity (%)** | <9% | <9% | <9% | <9% |
| **pH** | 7 | 9 | 9 | 10.5 |
| **Specific Surface (m²/gr)** | 1150 (m²/gr) | 1610 (m²/gr) | 1110 (m²/gr) | 1550 (m²/gr) |
| **Mesh Screen** | 8×30 | 8×30 | 8×30 | 8×30 |

percentage obtained is related to physical activated walnut carbon active and the lowest amount is related to chemical activated peach carbon active.

It also found that the obtained pH values for physical activated carbon are lower than chemical activated carbon in all cases. In other words, the chemical method of producing activated carbon provides a more alkaline environment, which is due to the use of NaOH alkaline substances. So that the pH value of chemical activated peach carbon active is equal to 9 and the pH value of physical activated peach carbon active is measured as 7. These values for walnut are equal to 10.5 and 9, respectively. Therefore, the highest pH obtained is related to chemical activated walnut carbon active and the lowest pH obtained is related to physical activated peach carbon active.

In this work, Scanning Electron Microscope (SEM- model JSM-IT800) was applied to examine the internal morphology of microstructures in micrometer and nanometer dimensions. To obtain more detailed information, the SEM images were investigated using ImageJ processing software. The parameters such as average wall thickness, average pore diameter, and the average pore density per unit of scanned area were calculated and evaluated. The analysis of the SEM images is presented in Table 2.

As can be found from Table 2, the behavior of pore density of the activated carbons is the same as the hardness. The pore density of the chemical activated peach carbon active is obtained as 81, which is the highest obtained pore density; and the pore density of the physical activated walnut carbon active has obtained as the lowest one.

It was also resulted from Table 2 that the pore diameter of physical activated carbon is lower than chemical activated carbon in all cases. So that the pore diameter of chemical activated peach carbon active is equal to 22.08 μm and the pore diameter of physical activated peach carbon active is equal to 20.42 μm. These values for walnut are obtained as 22.74 μm and 21.86 μm, respectively. Therefore, the lowest value of pore diameter obtained is related to physical activated peach carbon active and the highest pore diameter value is related to chemical activated walnut carbon active. The results show that the pore diameter values in different obtained activated carbons are inversely related to the density. So that as the density of an active carbon increases, the value of pore diameter values decreases, and vice versa. This result is obtained because as the size of the pores increases, the density decreases.

Fig 1 shows the pore size distribution of peach kernel activated carbon and walnut shell activated carbon using either chemical or physical methods. As can be seen from Fig 1, the pore size distribution of walnut shell activated carbon is much narrower than the peach kernel activated carbon, and this result is observed in both activated carbon fabrication methods, chemical or physical methods.

In order to check the quality of each of the produced adsorbents in the treatment of oily wastewater, a water/oil emulsion was first prepared with a certain amount of diesel, gasoline, used oil or engine lubricant. For the synthesis of the mentioned emulsion, first a certain

**Table 2. Characteristics of fabricated activated carbons from precursors of peach kernel and walnut shell obtained from statistical studies of scanning electron microscope (SEM) images.**

| Parameter | Peach Activated Carbon | | Walnut Activated Carbon | |
|---|---|---|---|---|
| | Physical Activation | Chemical Activation | Physical Activation | Chemical Activation |
| **Pore Diameter (μm)** | 20.42 μm | 22.08 μm | 21.86 μm | 22.74 μm |
| **Standard Deviation** | 6.01% | 5.87% | 6.81% | 6.19% |
| **Shell Thickness** | 3.75 μm | 3.81 | 3.83 μm | 3.79 |
| **Pore Density** | 67 | 81 | 53 | 64 |

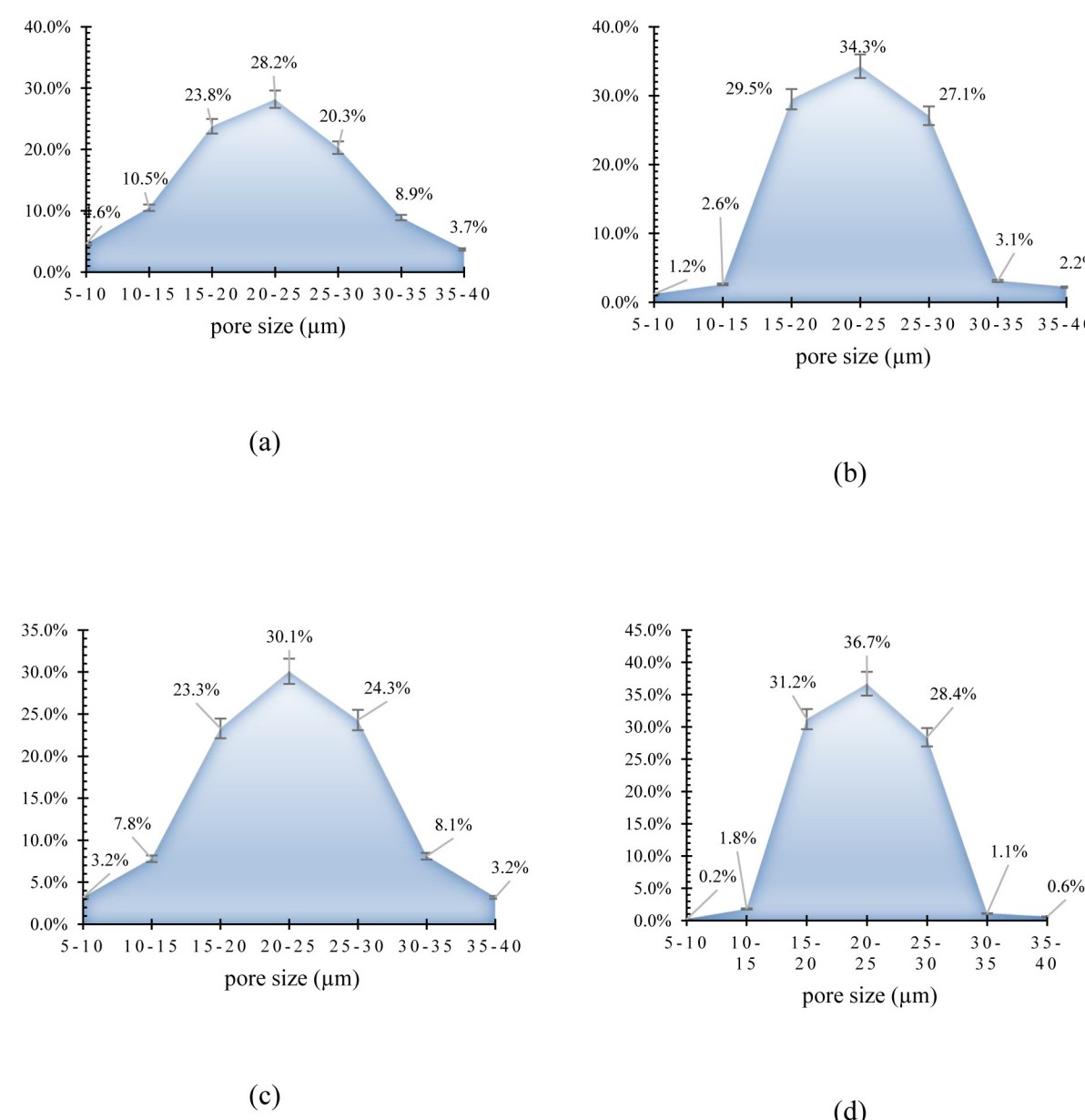

**Fig 1. The pore size distribution of a) physical peach kernel activated carbon, b) chemical peach kernel activated carbon, c) physical walnut shell activated carbon, d) chemical peach shell activated carbon.**

amount of the pollutant was poured into double distilled water and the desired emulsion was obtained using Tween80 as an emulsifier. The concentration of oil in all prepared feeds for different analysis is constant and is adjusted to 3000 mg/L (equivalent to 6400 ppm based on COD calculations). After adding Tween80 emulsifier to the system with a concentration of 100 mg/L, a high-speed homogenizer is applied to prepare an emulsion with high stability.

This homogenizing operation is performed for 30 minutes at 12,000 rpm. In order to determine the size distribution of oil particles in water, a DLS analysis (BeNano90) was performed. The synthesized oily wastewater particle sizes of diesel, gasoline, used oil, and engine lubricant

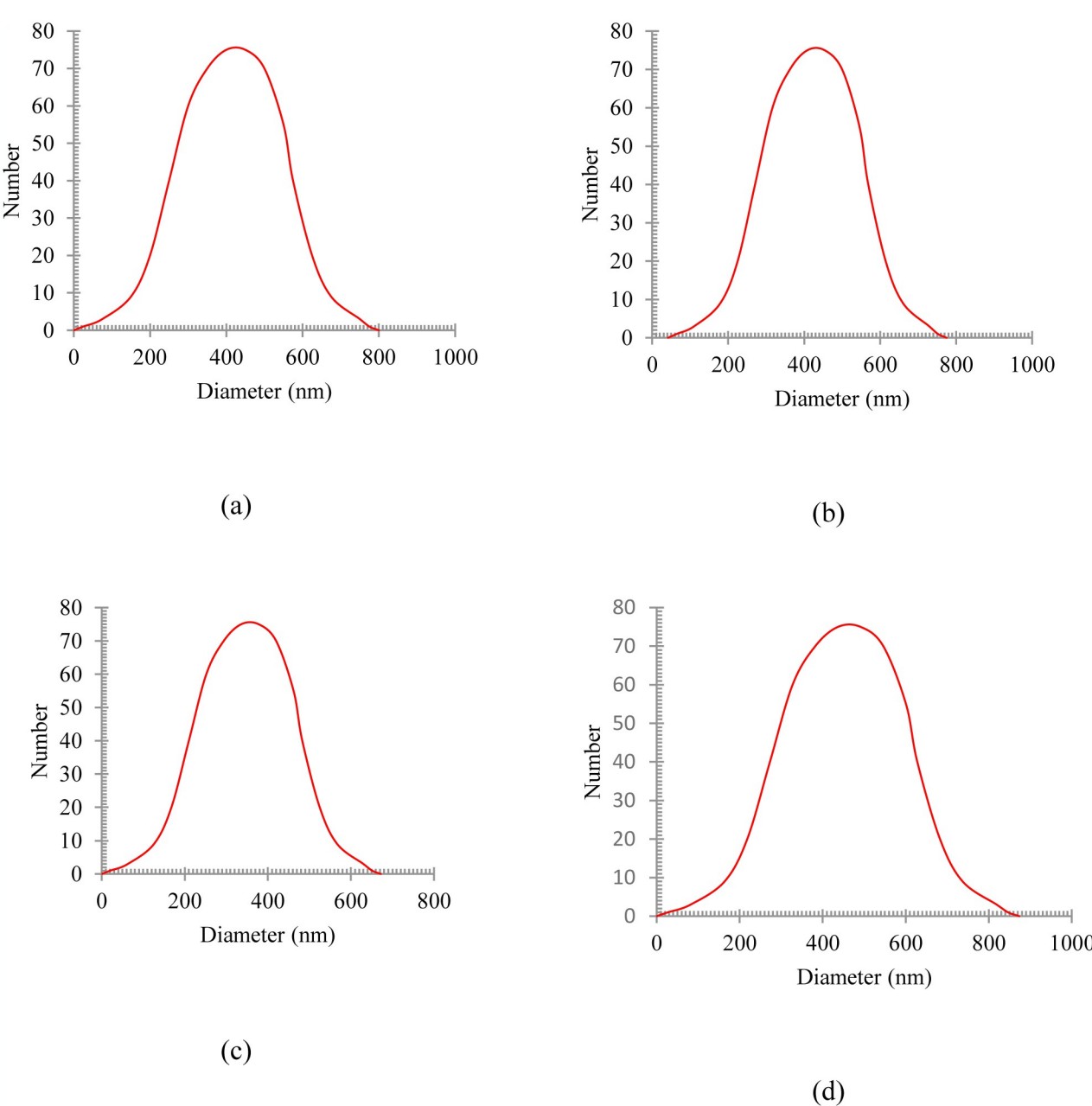

**Fig 2. The synthesized oily wastewater particle size of a) diesel, b) gasoline, c) used oil, and d) engine lubricant.**

are shown in Fig 2. As can be seen in Fig 2, the average particle sizes of these four different emulsions are in the range of 350 to 450 nm.

Fig 3 shows the temperature effects on the adsorption of different synthesized oily wastewater on physical activated peach carbon active. All cases show a decrease in adsorption when the temperature is increases, which can be referred to the exothermic reaction of the adsorption process.

Comparing the results of the adsorption process of diesel, gasoline, used oil, and engine lubricant on physical activated peach carbon active, which is presented in Fig 3, it can also be

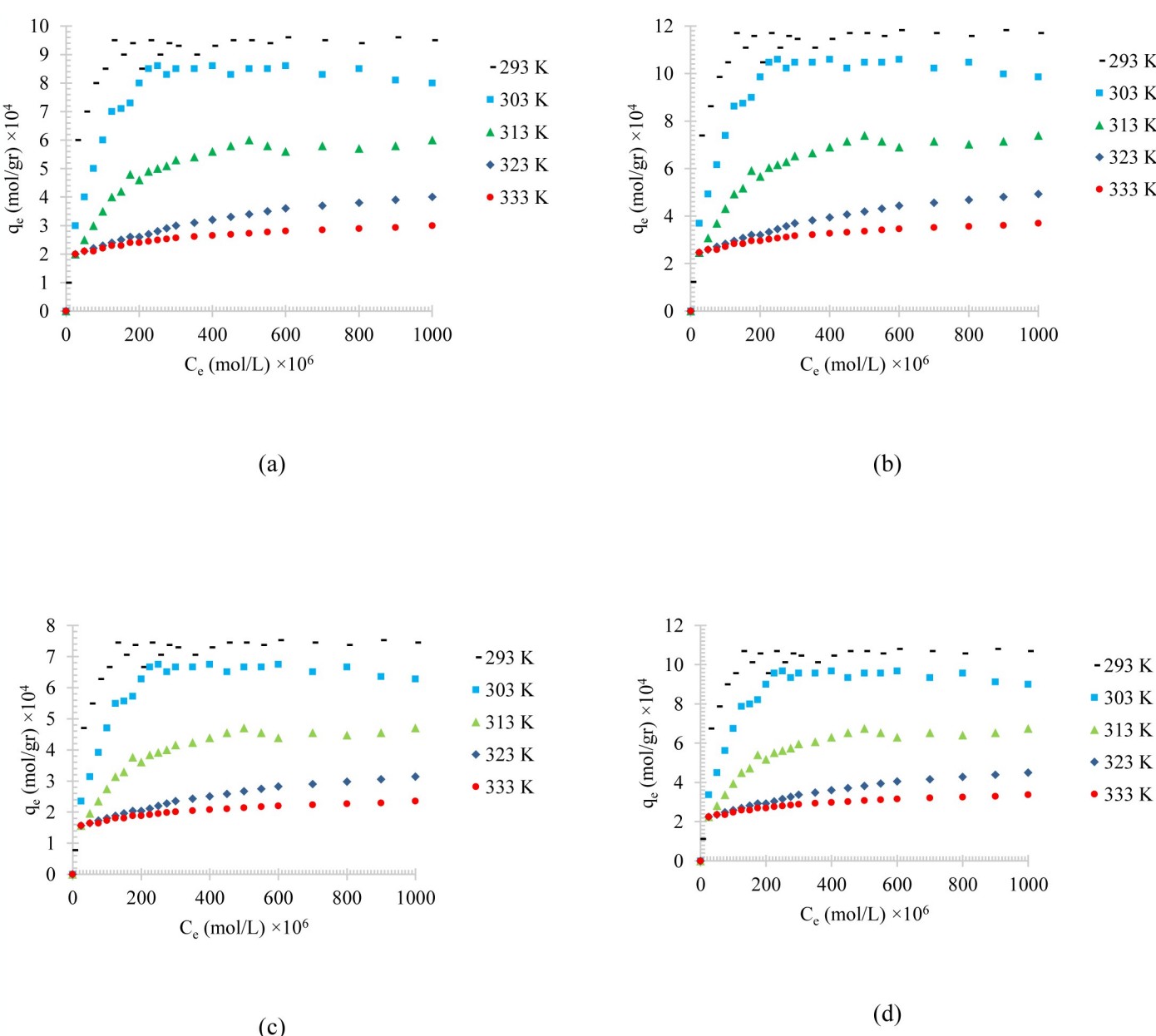

**Fig 3. The temperature effect on the adsorption of a) diesel, b) gasoline, c) used oil, and d) engine lubricant at different temperatures (293 to 333 K) and pH = 6 on physical activated peach carbon active.**

realized that, in general, the lower the density of the adsorbent material, the higher the equilibrium adsorption rate. Therefore, gasoline with a density of 0.73 kg/l showed the highest adsorption and used oil with a density of 0.93 kg/l showed the lowest equilibrium adsorption.

Figs 4–6, shows the temperature effects on the adsorption of different synthesized oily wastewater on chemical activated peach carbon active, physical activated walnut carbon active, and chemical activated walnut carbon active, respectively. As can be compared, in all cases, a decrease in adsorption can be seen with an increase in temperature.

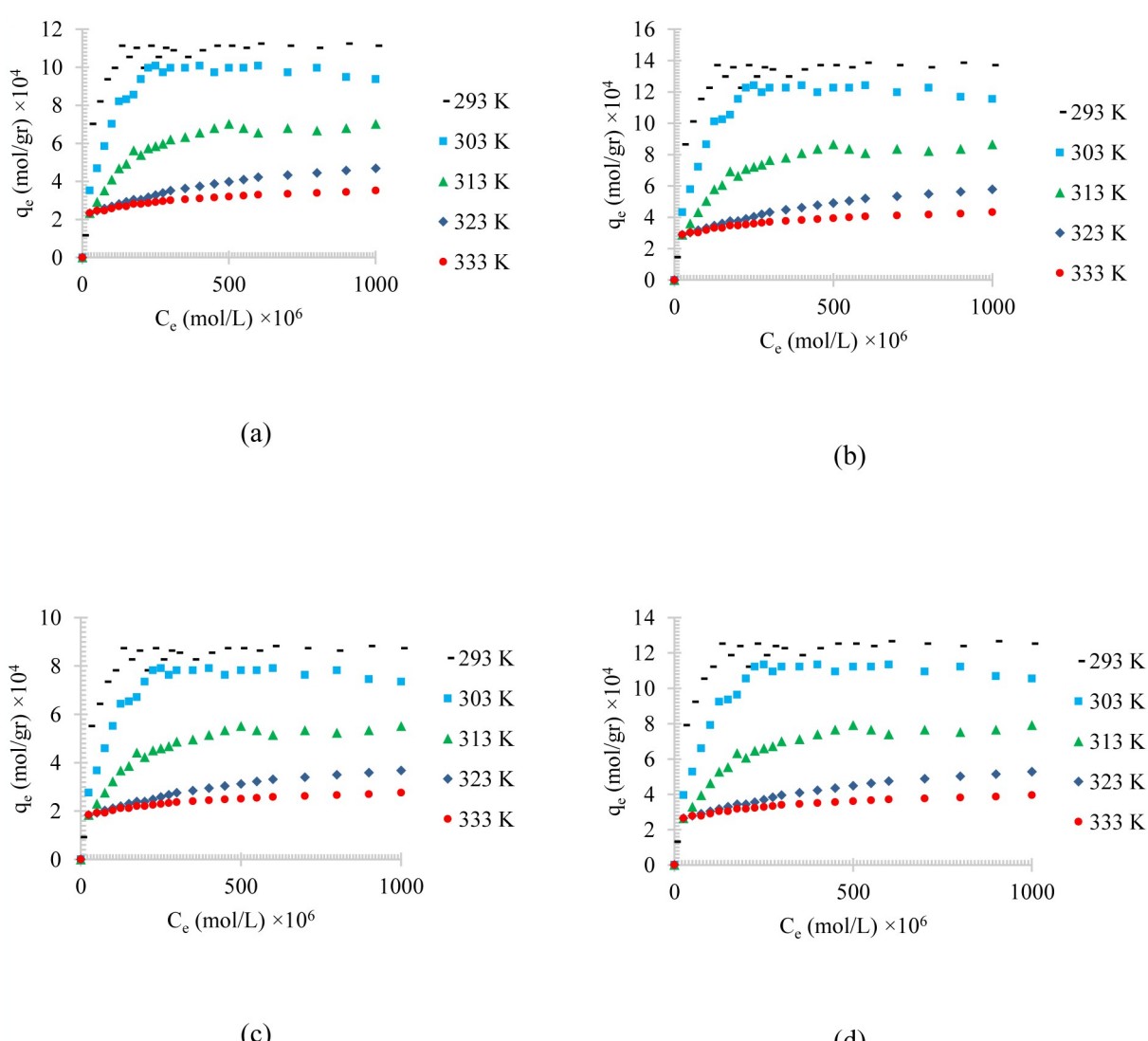

**Fig 4. The temperature effect on the adsorption of a) diesel, b) gasoline, c) used oil, and d) engine lubricant at different temperatures (293 to 333 K), and pH = 6, on chemical activated peach carbon active.**

In all of these figures (Figs 4–6), the same general relationship between density and maximum amount of adsorption is precise. In other words, comparing the results of the adsorption of diesel, gasoline, used oil, and engine lubricant on various types of synthesized activated carbon, shows that the lower the density of the adsorbed substance is, the higher the equilibrium adsorption rate is.

By comparing Figs 4–6, it can also be found that the highest amount of adsorption is in the chemical activated peach carbon active and the lowest amount of adsorption was obtained in the adsorption of the physical activated walnut carbon active. In other words, the peach activated carbon has a higher adsorption than walnut activated carbon, generally; and the adsorption amount can be increased using chemical methods; Therefore, the highest amount of adsorption is related to peach activated carbon that is made by chemical method, and the lowest amount of adsorption can be seen in walnut activated carbon that is made by physical

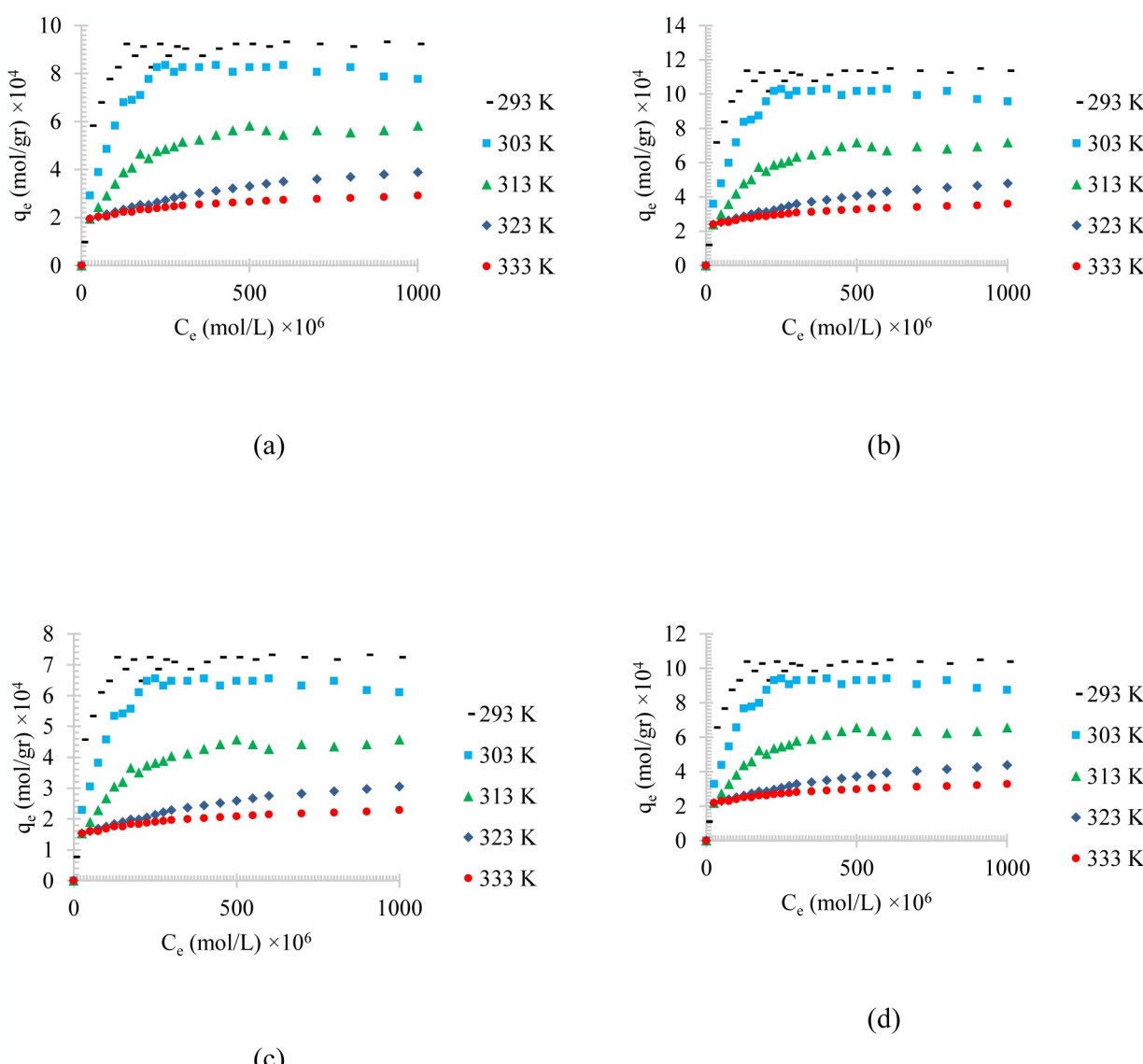

**Fig 5. The temperature effect on the adsorption of a) diesel, b) gasoline, c) used oil, and d) engine lubricant at different temperatures (293 to 333 K), and pH = 6, on physical activated walnut carbon active.**

method. This is due to the more suitable pore structure, more surface energy and more susceptible functional groups of peach-based carbon actives.

Fig 7 shows the pH effect on the adsorption of different synthesized oily wastewater on physical activated peach carbon active. As can be seen from Fig 7 a pH increase causes an increase in the adsorption. Comparing the results of the adsorption process of diesel, gasoline, used oil, and engine lubricant on physical activated peach carbon active, which is presented in Fig 7, it can also be realized that, in general, the lower the density of the adsorbent material, the higher the equilibrium adsorption rate. Therefore, gasoline with a density of 0.73 kg/l showed the highest adsorption and used oil with a density of 0.93 kg/l showed the lowest equilibrium adsorption, which is similar to the Fig 3.

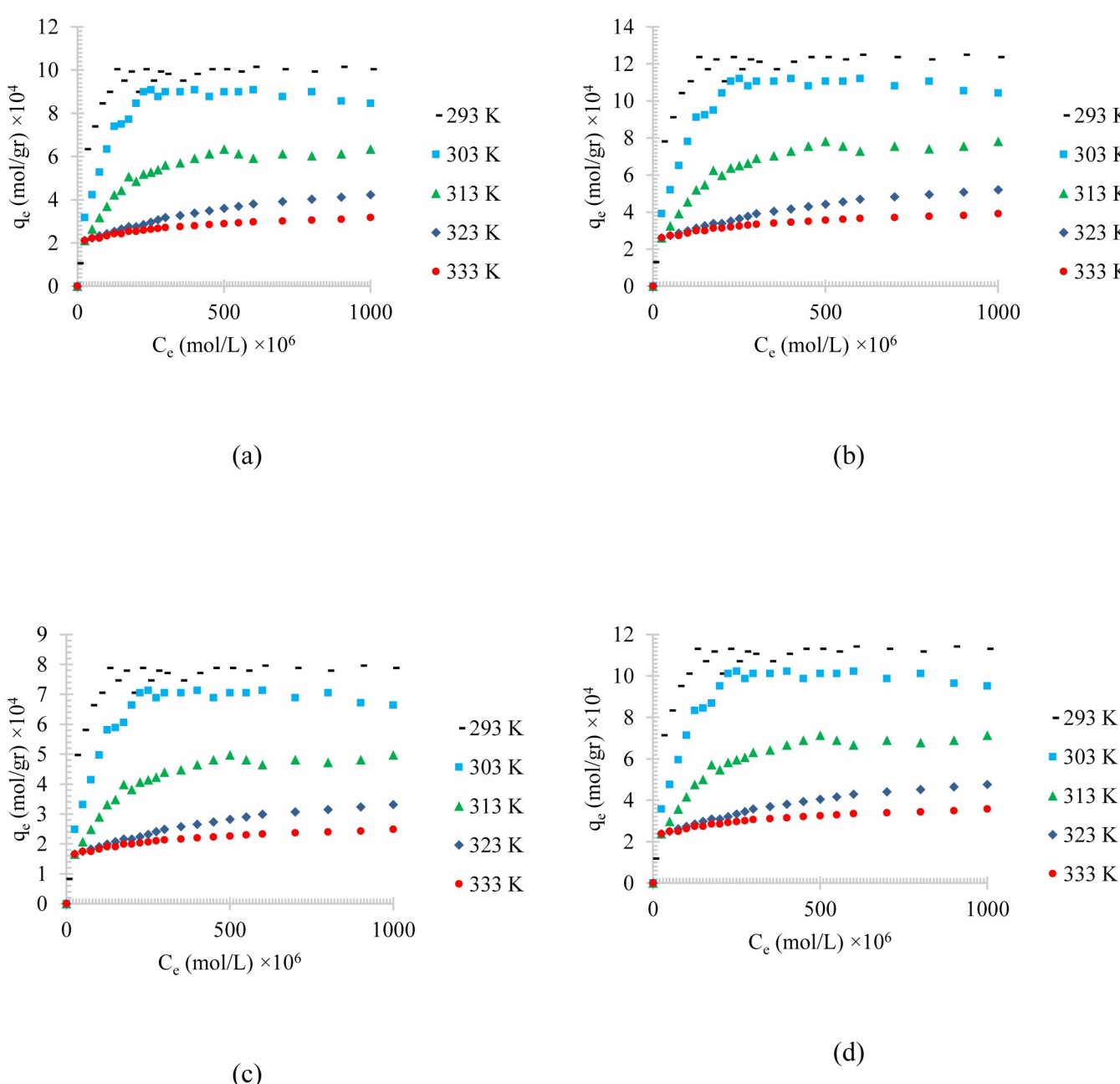

(a)

(b)

(c)

(d)

**Fig 6. The temperature effect on the adsorption of a) diesel, b) gasoline, c) used oil, and d) engine lubricant at different temperatures (293 to 333 K), and pH = 6, on chemical activated walnut carbon active.**

Figs 8–10, shows the pH value effects on the adsorption of different synthesized oily wastewater on chemical activated peach carbon active, physical activated walnut carbon active, and chemical activated walnut carbon active, respectively. As can be compared, in all cases, an increase in adsorption can be seen with an increase in pH.

In all of these figures (Figs 4–6), the same general relationship between density and maximum amount of adsorption is precise. In other words, comparing the results of the adsorption

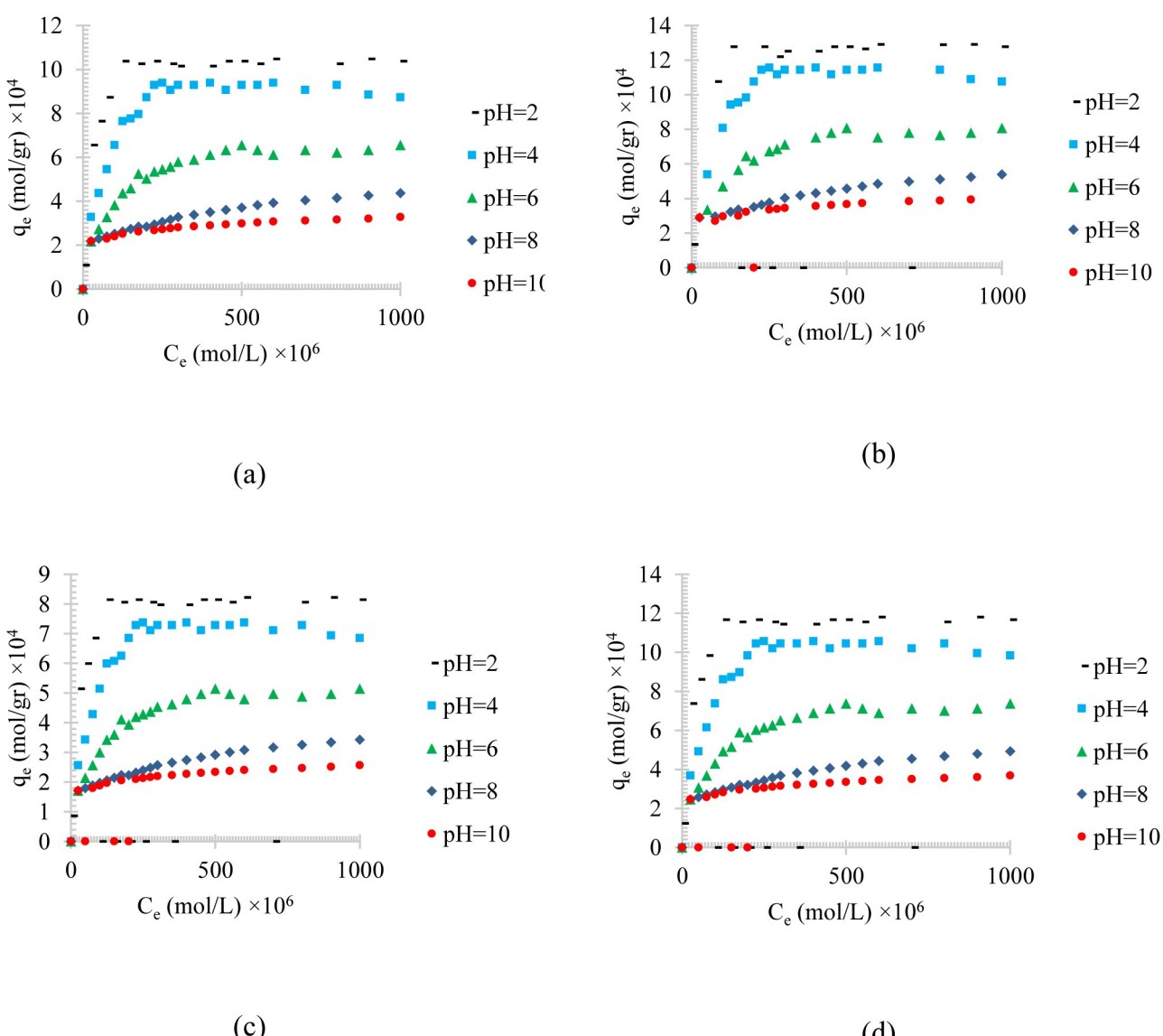

(a)

(b)

(c)

(d)

**Fig 7. The pH effect on the adsorption of a) diesel, b) gasoline, c) used oil, and d) engine lubricant at different pH (2 to 10), and T = 313 K, on physical activated peach carbon active.**

of diesel, gasoline, used oil, and engine lubricant on various types of synthesized activated carbon, shows that the lower the density of the adsorbed substance is, the higher the equilibrium adsorption rate is, in all pH values.

By comparing Figs 8–10, it can also be found that the highest amount of adsorption was obtained in the chemical activated peach carbon active and the lowest amount of adsorption was obtained in the adsorption of the physical activated walnut carbon active. In other words, the peach activated carbon has a higher adsorption than walnut activated carbon, generally; and the adsorption amount can be increased using chemical methods; Therefore, the highest amount of adsorption is related to peach activated carbon that is made by chemical method, and the lowest amount of adsorption can be seen in walnut activated carbon that is made by physical method.

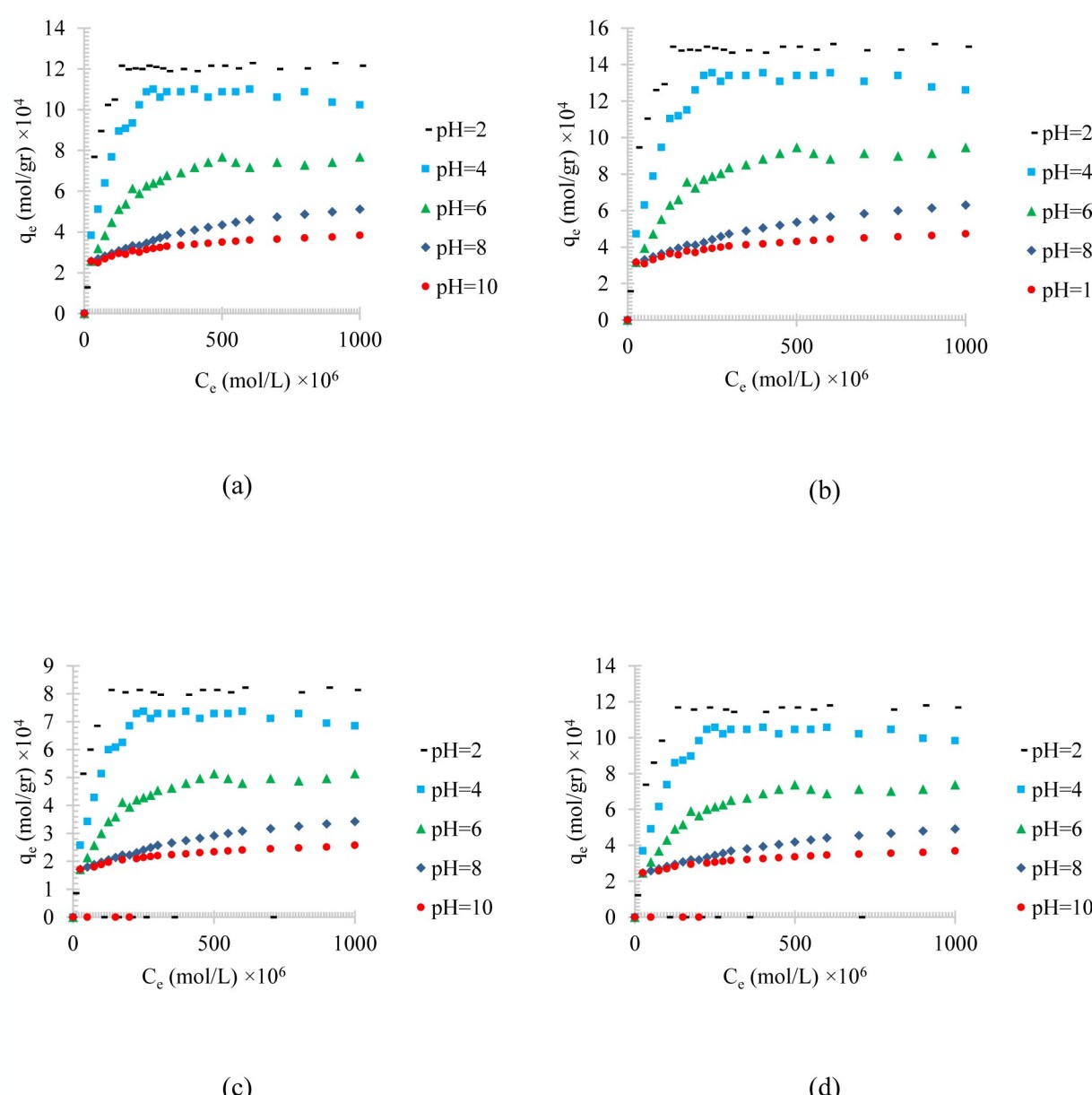

**Fig 8. The pH effect on the adsorption of a) diesel, b) gasoline, c) used oil, and d) engine lubricant at different pH (2 to 10), and T = 313 K, on chemical activated peach carbon active.**

The temperature effect on the linearized model of Langmuir isotherm adsorption of diesel, gasoline, used oil, and engine lubricant at different temperatures on the physical activated peach carbon active is shown Fig 11. Moreover, the pH effect on the linearized model of Langmuir isotherm adsorption of diesel, gasoline, used oil, and engine lubricant at different pH on the physical activated peach carbon active is shown Fig 12.

To analyze the compatibility of adsorption isotherms with experimental data and to predict the adsorption process, the adsorbed values were calculated using different isotherms. To select the most appropriate isotherm, the least square method has been used using the value of $R^2$ or correlation coefficient.

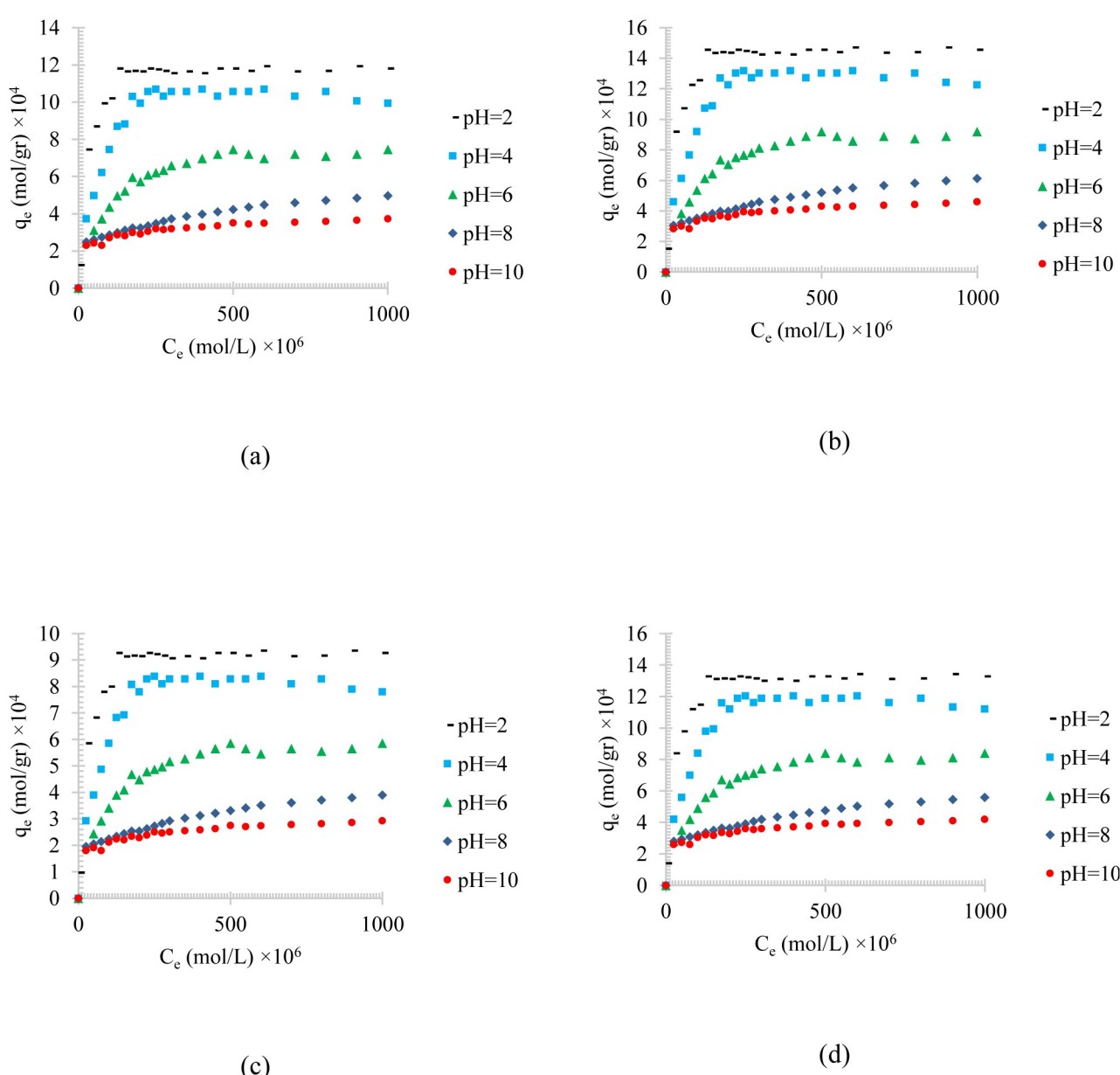

**Fig 9. The pH effect on the adsorption of a) diesel, b) gasoline, c) used oil, and d) engine lubricant at different pH (2 to 10), and T = 313 K, on physical activated walnut carbon active.**

Three different isotherm models of adsorption process, named Langmuir, Temkin, and Freundlich isotherms, were used to corelate the adsorption process, and the isotherm parameters were determined through the linearized experimental data. The isotherm parameters are listed in Table 3.

Since the Langmuir isotherm model had the best correlation in all cases compared to the Freundlich and Temkin isotherms, based on the correlation coefficient, and the calculated $R^2$ values were greater than 0.99, the adsorption of gasoline, used oil, and engine lubricant using different carbon actives were only correlated by Langmuir model. Therefore, the Langmuir

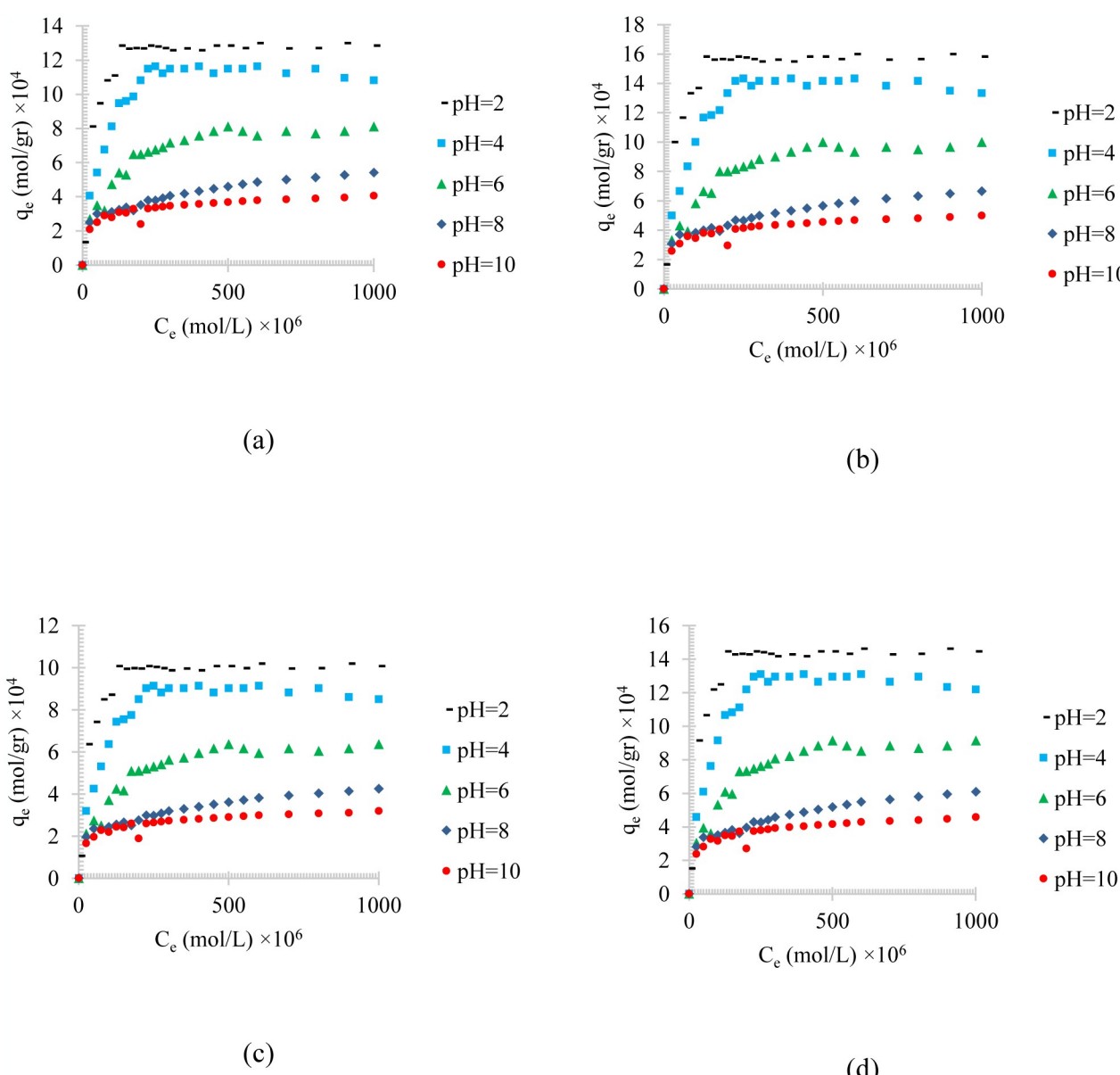

**Fig 10. The pH effect on the adsorption of a) diesel, b) gasoline, c) used oil, and d) engine lubricant at different pH (2 to 10), and T = 313 K, on chemical activated walnut carbon active.**

parameters of gasoline, used oil, and engine lubricant using chemical activated peach carbon active at different temperatures (293 to 333 K), and pH = 6 is shown in Table 4.

The Langmuir parameters of diesel, gasoline, used oil, and engine lubricant using chemical activated peach carbon active, at different temperatures is shown in Table 5. The same for the physical and chemical activated walnut carbon active are listed in Tables 6 and 7.

## 4. Conclusions

Today, oil spills into the sea and creating oil pollution on the water surface have become one of the biggest global concerns that have caused environmental problems. It is worth noting

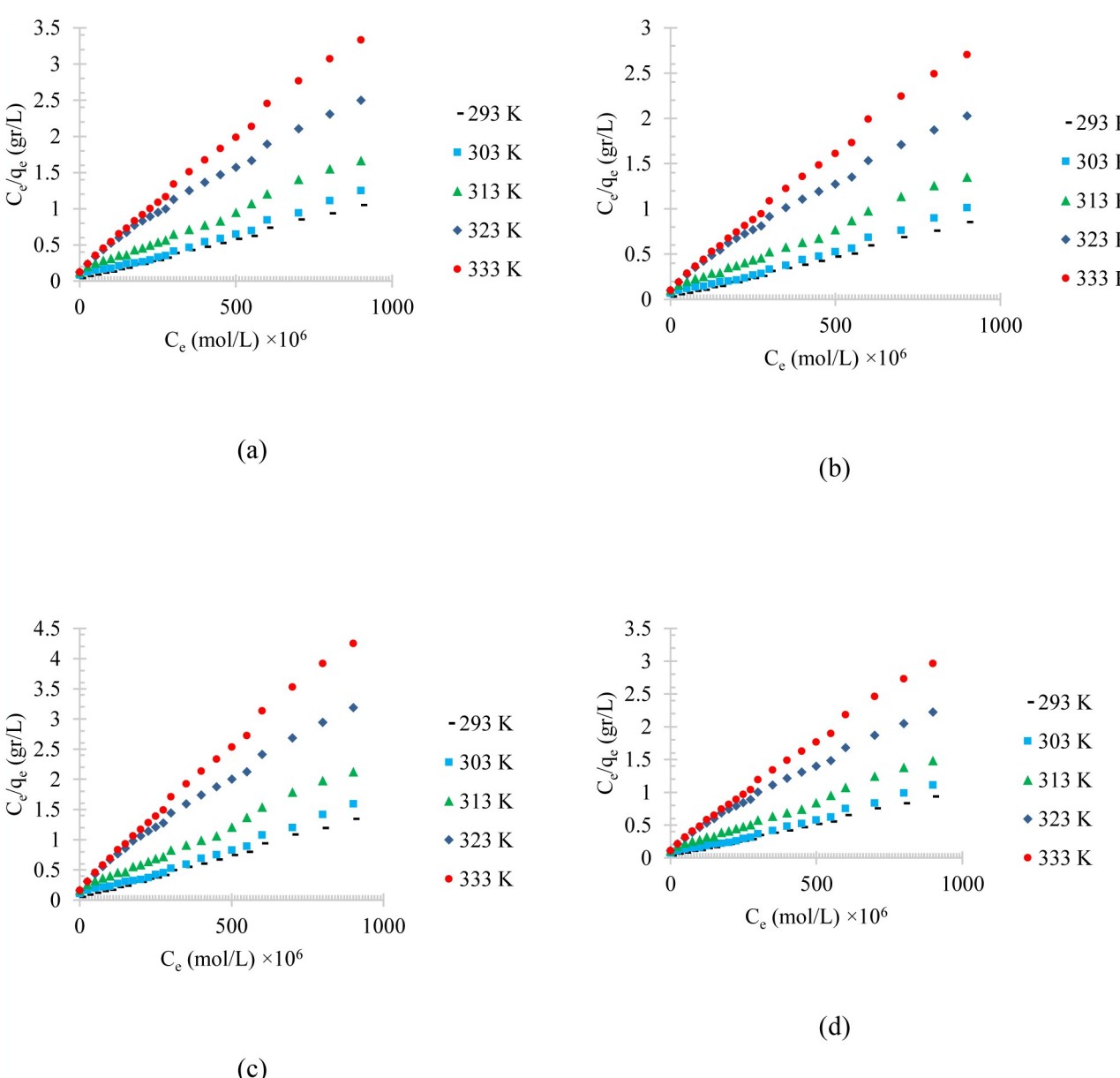

(a)

(b)

(c)

(d)

**Fig 11. The temperature effect on the linearized model of Langmuir isotherm adsorption of a) diesel, b) gasoline, c) used oil, and d) engine lubricant at different temperatures (293 to 333 K), and pH = 6, on physical activated peach carbon active.**

that solving this environmental problem by itself has also caused many economic problems. So far, many methods have been used to remove this type of pollution, including chemical and physical methods. The adsorption method is a physical method that can be suitable and cheap according to the choice of adsorbent. In recent years, the use of adsorbents, especially agricultural wastes, has attracted the attention of many researchers. Despite the many articles about activated carbon with different precursors, no in-depth research has been carried out to understand the causes of the difference in surface adsorption characteristics of activated carbon with different precursors and different activation processes. In this work, four different types of activated carbon were fabricated using peach kernel or walnut shell during chemical and

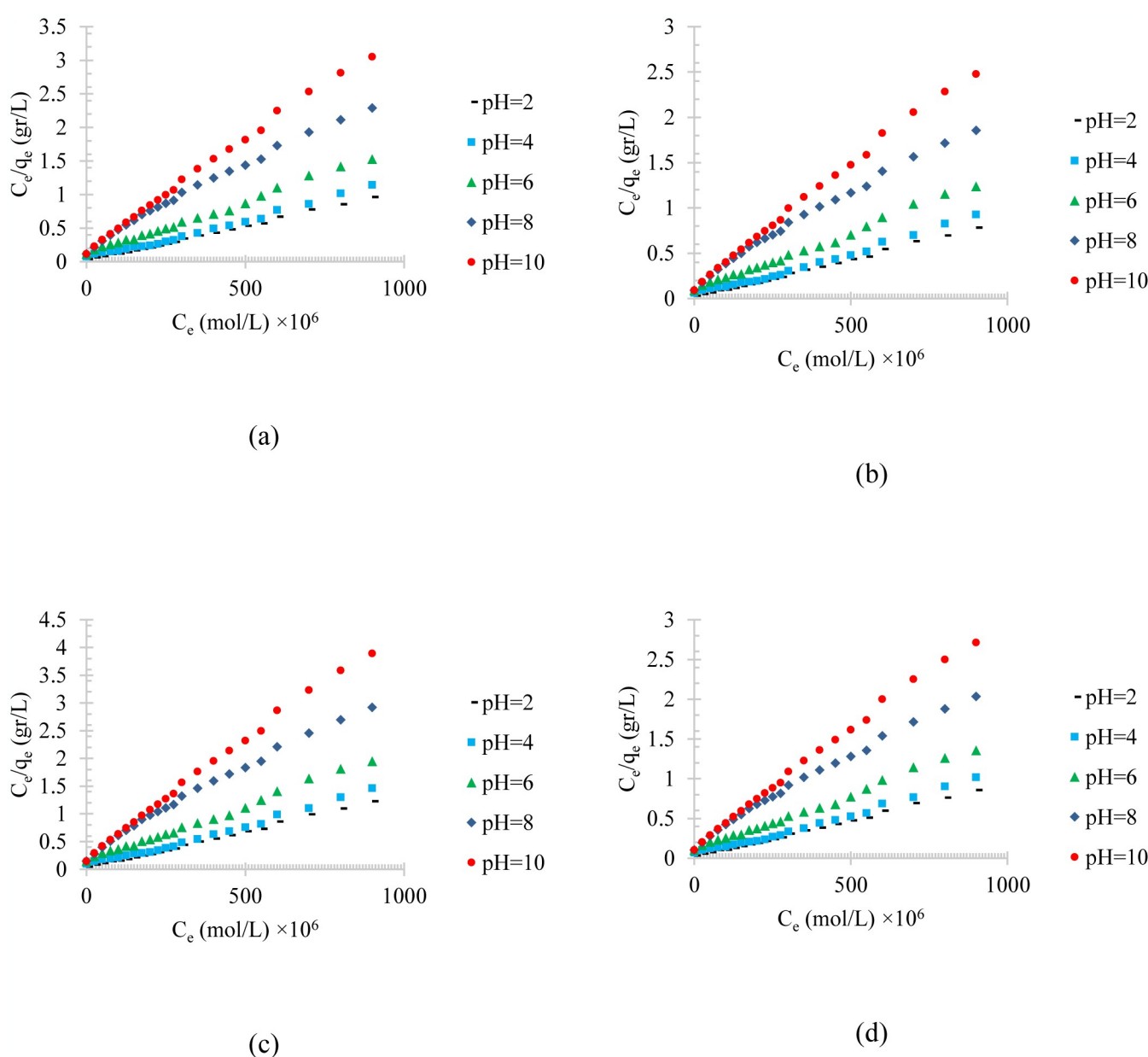

**Fig 12. The pH effect on the linearized model of Langmuir isotherm adsorption of a) diesel, b) gasoline, c) used oil, and d) engine lubricant at different pH (2 to 10), and T = 313 K, on physical activated peach carbon active.**

physical methods; and the adsorption capacity of the fabricated carbon actives to treatment of oily wastewater including diesel, gasoline, used oil and engine lubricant was investigated. The results show that the lower pollution density has the higher adsorbed amount. The temperature and pH effects on the adsorption of different synthesized oily wastewater was studied and it was found that a decrease in adsorption can be seen with an increase in temperature or decreasing the pH value, which can be referred to this fact that the adsorption is an exothermic process. Finally, to analyze the compatibility of adsorption isotherms with experimental data and to predict the adsorption process, three different isotherms named Langmuir, Temkin,

**Table 3. Parameters of different isotherms for diesel removal using physical activated peach carbon active at different temperatures (293 to 333 K), and pH = 6.**

| Model | Parameters | 293 K | 303 K | 313 K | 323 K | 333 K |
|---|---|---|---|---|---|---|
| Langmuir | $K_l \times 10^{-4}$ | 7.678 | 4.416 | 1.490 | 0.998 | 2.189 |
| | $q_m \times 10^4$ | 9.66 | 8.577 | 6.360 | 4.224 | 3.046 |
| | $R^2$ | 0.999 | 0.994 | 0.997 | 0.989 | 0.998 |
| Freundlich | $K_F \times 10^{-2}$ | 4.869 | 1.823 | 1.775 | 6.25 | 15.221 |
| | $N$ | 10.2 | 4.01 | 3.42 | 4.56 | 7.79 |
| | $R^2$ | 0.712 | 0.722 | 0.912 | 0.923 | 0.974 |
| Temkin | $K_T \times 10^{-8}$ | 4.229 | 2.050 | 0.009 | 0.025 | 0.109 |
| | $b \times 10^{-7}$ | 2.988 | 0.361 | 2.442 | 4.123 | 10.260 |
| | $R^2$ | 0.692 | 0.752 | 0.912 | 0.922 | 0.947 |

**Table 4. Langmuir parameters of gasoline, used oil, and engine lubricant using physical activated peach carbon active at different temperatures (293 to 333 K), and pH = 6.**

| Model | Parameters | 293 K | 303 K | 313 K | 323 K | 333 K |
|---|---|---|---|---|---|---|
| Gasoline | $K_l \times 10^{-4}$ | 7.633 | 4.401 | 1.490 | 0.998 | 0.998 |
| | $q_m \times 10^4$ | 11.909 | 10.568 | 7.836 | 5.204 | 5.204 |
| | $R^2$ | 0.999 | 0.994 | 0.997 | 0.990 | 0.997 |
| Used Oil | $K_l \times 10^{-4}$ | 7.667 | 4.409 | 1.490 | 0.998 | 1.189 |
| | $q_m \times 10^4$ | 7.582 | 6.729 | 4.989 | 3.313 | 2.39 |
| | $R^2$ | 0.999 | 0.993 | 0.997 | 0.989 | 0.998 |
| Engine Lubricant | $K_l \times 10^{-4}$ | 7.649 | 4.407 | 1.490 | 0.998 | 2.189 |
| | $q_m \times 10^4$ | 10.879 | 9.654 | 7.159 | 4.755 | 3.429 |
| | $R^2$ | 0.999 | 0.994 | 0.997 | 0.989 | 0.997 |

**Table 5. Langmuir parameters of diesel, gasoline, used oil, and engine lubricant using chemical activated peach carbon active at different temperatures (293 to 333 K), and pH = 6.**

| Model | Parameters | 293 K | 303 K | 313 K | 323 K | 333 K |
|---|---|---|---|---|---|---|
| Diesel | $K_l \times 10^{-4}$ | 7.680 | 4.403 | 1.498 | 0.998 | 2.189 |
| | $q_m \times 10^4$ | 11.323 | 10.047 | 7.450 | 4.948 | 3.568 |
| | $R^2$ | 0.999 | 0.994 | 0.997 | 0.989 | 0.997 |
| Gasoline | $K_l \times 10^{-4}$ | 7.625 | 43.76 | 1.490 | 0.990 | 2.188 |
| | $q_m \times 10^4$ | 13.950 | 1.248 | 9.180 | 6.096 | 4.397 |
| | $R^2$ | 0.999 | 0.993 | 0.997 | 0.989 | 0.996 |
| Used Oil | $K_l \times 10^{-4}$ | 7.658 | 4.405 | 1.490 | 1.001 | 2.189 |
| | $q_m \times 10^4$ | 8.882 | 7.882 | 5.844 | 3.881 | 2.800 |
| | $R^2$ | 0.998 | 0.994 | 0.997 | 0.986 | 0.998 |
| Engine Lubricant | $K_l \times 10^{-4}$ | 7.692 | 4.420 | 1.491 | 0.999 | 2.189 |
| | $q_m \times 10^4$ | 12.745 | 11.309 | 8.386 | 5.569 | 4.017 |
| | $R^2$ | 0.998 | 0.994 | 0.997 | 0.990 | 0.997 |

**Table 6. Langmuir parameters of diesel, gasoline, used oil, and engine lubricant using physical activated walnut carbon active at different temperatures (293 to 333 K), and pH = 6.**

| Model | Parameters | 293 K | 303 K | 313 K | 323 K | 333 K |
|---|---|---|---|---|---|---|
| Diesel | $K_l \times 10^{-4}$ | 7.662 | 4.412 | 1.490 | 0.998 | 2.189 |
| | $q_m \times 10^4$ | 9.389 | 8.332 | 6.178 | 4.103 | 2.959 |
| | $R^2$ | 0.999 | 0.994 | 0.997 | 0.989 | 0.997 |
| Gasoline | $K_l \times 10^{-4}$ | 7.649 | 4.407 | 1.489 | 0.997 | 2.180 |
| | $q_m \times 10^4$ | 11.568 | 10.266 | 7.612 | 5.055 | 3.646 |
| | $R^2$ | 0.999 | 0.993 | 0.995 | 0.990 | 0.996 |
| Used Oil | $K_l \times 10^{-4}$ | 7.670 | 4.409 | 1.487 | 0.944 | 2.189 |
| | $q_m \times 10^4$ | 7.365 | 6.536 | 4.847 | 3.219 | 2.321 |
| | $R^2$ | 0.999 | 0.989 | 0.997 | 0.989 | 0.998 |
| Engine Lubricant | $K_l \times 10^{-4}$ | 7.630 | 4.405 | 1.490 | 0.998 | 2.189 |
| | $q_m \times 10^4$ | 10.569 | 9.378 | 6.954 | 4.618 | 3.331 |
| | $R^2$ | 0.999 | 0.990 | 0.983 | 0.990 | 0.998 |

**Table 7. Langmuir parameters of diesel, gasoline, used oil, and engine lubricant using chemical activated walnut carbon active at different temperatures (293 to 333 K), and pH = 6.**

| Model | Parameters | 293 K | 303 K | 313 K | 323 K | 333 K |
|---|---|---|---|---|---|---|
| Diesel | $K_l \times 10^{-4}$ | 7.646 | 4.411 | 1.490 | 0.998 | 2.189 |
| | $q_m \times 10^4$ | 10.217 | 9.067 | 6.723 | 4.465 | 3.220 |
| | $R^2$ | 0.999 | 0.993 | 0.997 | 0.989 | 0.997 |
| Gasoline | $K_l \times 10^{-4}$ | 7.637 | 4.409 | 1.490 | 0.998 | 2.189 |
| | $q_m \times 10^4$ | 12.589 | 11.172 | 8.284 | 5.501 | 3.968 |
| | $R^2$ | 0.999 | 0.998 | 0.997 | 0.999 | 0.999 |
| Used Oil | $K_l \times 10^{-4}$ | 7.653 | 4.408 | 1.491 | 1.009 | 2.190 |
| | $q_m \times 10^4$ | 8.016 | 7.111 | 5.274 | 3.502 | 2.526 |
| | $R^2$ | 0.999 | 0.992 | 0.999 | 0.999 | 0.998 |
| Engine Lubricant | $K_l \times 10^{-4}$ | 7.630 | 4.413 | 1.489 | 1.025 | 2.189 |
| | $q_m \times 10^4$ | 11.501 | 10.206 | 7.569 | 5.026 | 3.625 |
| | $R^2$ | 0.999 | 0.999 | 0.998 | 0.999 | 0.998 |

and Freundlich isotherms were applied and their parameters were correlated. The correlation results show that the Langmuir isotherm had the best correlation in all cases compared to the Freundlich and Temkin isotherms, based on the correlation coefficient, and the calculated $R^2$ values. The characteristics of fabricated activated carbons from precursors of peach kernel and walnut shell using physical or chemical method were investigated, and the iodine number, hardness, pH, density, specific surface, pore density, ash and humidity percentage were reported. It was found that peach activated carbon has a higher iodine number than walnut activated carbon, and this amount can be increased using chemical methods; therefore, the highest amount of iodine number is related to peach activated carbon that is made by chemical method, and the lowest amount of iodine number is seen in walnut activated carbon that is made by physical method. This phenomenon can also be seen in hardness and specific surface area, so that the chemical activated peach carbon active has provided the highest hardness and physical activated walnut carbon active has obtained the lowest hardness value. The pore

diameter of physical activated carbon is lower than chemical activated carbon and the density of physical activated carbon is higher than chemical activated carbon in all cases. Furthermore, the chemical method of producing activated carbon provides a more alkaline environment, which is due to the use of NaOH alkaline substances. It is suggested that in the continuation of this research, the kinetics of absorption in each of the fabricated activated carbons also be investigated and the relationship between the kinetics and thermodynamics of the absorption process be studied and discussed.

## Acknowledgments

The authors acknowledge the anonymous reviewers for their valuable suggestions that helped to improve the quality of the manuscript.

## Author Contributions

**Conceptualization:** Atef El Jery, Nadhir Al-Ansari, Miklas Scholz.

**Data curation:** Atef El Jery.

**Formal analysis:** Khaled Mohamed Khedher.

**Investigation:** Khaled Mohamed Khedher.

**Methodology:** Khaled Mohamed Khedher.

**Project administration:** Nadhir Al-Ansari.

**Resources:** Hayder Mahmood Salman, Nadhir Al-Ansari.

**Software:** Hayder Mahmood Salman, Nadhir Al-Ansari.

**Validation:** Saad Sh. Sammen, Miklas Scholz.

**Visualization:** Saad Sh. Sammen, Miklas Scholz.

**Writing – original draft:** Hayder Mahmood Salman, Saad Sh. Sammen.

**Writing – review & editing:** Saad Sh. Sammen.

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
