## [Decision Letter · Decision Letter 0]

30 Mar 2023

PONE-D-23-04431Thermodynamic and Structural Investigation of Oily Wastewater Treatment Using Peach Kernel and Walnut Shell Based Activated CarbonPLOS ONE

Dear Dr. Al-Ansari,

Thank you for submitting your manuscript to PLOS ONE. After careful consideration, we feel that it has merit but does not fully meet PLOS ONE’s publication criteria as it currently stands. Therefore, we invite you to submit a revised version of the manuscript that addresses the points raised during the review process.

We look forward to receiving your revised manuscript.

Kind regards,

Nor Adilla Rashidi, Ph.D.

Academic Editor

PLOS ONE

Journal Requirements:

"This work was supported by the King Khalid University, Abha, Saudi Arabia (by grant number R.G.P. 2/57/44). We express our gratitude to the Deanship of Scientific Research, King Khalid University, for its support of this study."

"This research work was supported by the Deanship of Scientific Research at King Khalid University under the grant number GRP.2/57/44

Reviewers' comments:

Reviewer's Responses to Questions

**Comments to the Author**

1. Is the manuscript technically sound, and do the data support the conclusions?

Reviewer #1: Partly

Reviewer #2: Partly

2. Has the statistical analysis been performed appropriately and rigorously? 

Reviewer #1: I Don't Know

Reviewer #2: N/A

3. Have the authors made all data underlying the findings in their manuscript fully available?

Reviewer #1: Yes

Reviewer #2: No

4. Is the manuscript presented in an intelligible fashion and written in standard English?

Reviewer #1: Yes

Reviewer #2: Yes

5. Review Comments to the Author

Reviewer #1: • Abstract required further refinement to ensure better understandings (i.e Line 28-30, line 38-40) and contained grammatical errors. Need to proofread the manuscript before submitting.

• Introduction part need to be fully revised especially the continuity in sentences and between paragraphs. Some of the statements are too broad.

• Justify the statement on chemical activation process (Line 93-96). There are processes that involve two-step synthesis as well. Some chemical activation uses cost-effective activated carbon production; hence the statement might be invalid. Please justify further.

• If there is no-in depth research on said topic, what did most people research and review on? Strengthen the novelty.

• “FTIR and XRD analyzes and SEM images have been used to identify the functional groups of the synthesized activated carbon surface, and to measure the pore structure of the synthesized carbon active” Which characterization is for which, revise the sentence for better clarity.

• In section 2.2.2, author mentioned on chemical activating agent of phosphoric acid and ZnCl2, but at the end didn’t conclude which one is being used in the methodology part. Amend the section accordingly.

• The section related to synthesis need to be revised with the addition of comparison of previous work in table form.

• Should provide surface morphology images (SEM) to strengthen the activated carbon properties rather than just put in table.

• More references need especially comparing the result in discussion.

Reviewer #2: 1. BET data should be supplemented.

2. The authors have performed the measurement at different temperatures. The authors may need to perform the required thermodynamic analysis, since the title has been stated so.

3. How does "no-in depth" research on the cause of the difference in surface adsorption characteristics help in this study?

4. Based on "3", will the analysis be different if the method of activation is changed?

5. Formatting issue in Table 2.

6. How do the performance of the studied adsorbents reported as compared to the literature studies?

6. PLOS authors have the option to publish the peer review history of their article (what does this mean?). If published, this will include your full peer review and any attached files.

Reviewer #1: No

Reviewer #2: No

---

## [Author Response · Author response to Decision Letter 0]

17 May 2023

Response to the Reviewers

The authors would like to thank the editor and the reviewers for their thoughtful comments and efforts toward improving our manuscript. A point-by-point response to the comments of the reviewers is as follows. 

Reviewer 1

Comment 1: Abstract required further refinement to ensure better understandings (i.e Line 28-30, line 38-40) and contained grammatical errors. Need to proofread the manuscript before submitting.

Answer: 

Thank you very much for your comment. The comment was applied and the Abstract section was further refined to ensure better understandings which is highlighted (Line 26-29 and Line 36-40). Furthermore, the manuscript was checked grammatically and was edited properly. The added text is as follow:

“This present study found that the peach activated carbon with a chemical process had the highest hardness (97%) compared to other types of activated carbon and the physical activated walnut carbon active has obtained the lowest hardness value (87%).”

“Furthermore, the study investigated the impact of temperature and pH on the adsorption of various synthesized oily wastewater. The findings revealed that an increase in temperature led to a decrease in adsorption capacity, while the adsorption rate increased with increasing temperature. Moreover, it was found that a decrease in adsorption observed when the adsorbent is negatively charged and the pH is lowered”

Comment 2: Introduction part need to be fully revised especially the continuity in sentences and between paragraphs. Some of the statements are too broad.

Answer: 

Thank you very much for your comment. The comment was applied and the Introduction part was revised and rewritten to ensure better understandings which is highlighted in the manuscript. Please see Introduction section and the highlighted text.

Comment 3: Justify the statement on chemical activation process (Line 93-96). There are processes that involve two-step synthesis as well. Some chemical activation uses cost-effective activated carbon production; hence the statement might be invalid. Please justify further.

Answer: 

Thank you for your thoughtfulness. Yes, you are right, some chemical activation uses cost-effective activated carbon production. But in general, it can be said that the chemical method uses more expensive materials; although there are some exceptions. The statement was edited as follows (Line 91-93):

“Generally, the chemical method uses more expensive materials and on a smaller scale than the physical method, however, some chemical activation uses cost-effective activated carbon production”

Comment 4: If there is no-in depth research on said topic, what did most people research and review on? Strengthen the novelty.

Answer: 

Thank you very much for your comment. It is important to note that the lack of in-depth research on a topic does not necessarily mean that there has been no research carried out on an area of study. It may indicate the need for further research and exploration in this area. In other words, this sentence means that the results of the works of other researchers should be placed next to each other and by covering the research gaps, a new and original result should emerge in order to open new horizons of science. Exactly what this work has tried to do to understand the causes of the difference in surface adsorption characteristics of the obtained carbon active with different precursors and different activation processes. However, in order to avoid misunderstanding, it was changed as below:

“Despite the numerous articles on activated carbon with different precursors in the adsorption process, more extensive research is needed to understand the causes of differences in the surface adsorption characteristics of activated carbon with different precursors and different activation processes.”

Comment 5: “FTIR and XRD analyzes and SEM images have been used to identify the functional groups of the synthesized activated carbon surface, and to measure the pore structure of the synthesized carbon active” Which characterization is for which, revise the sentence for better clarity.

Answer: 

Thank you very much for your comment. The comment was applied and the sentence was revised as follows for better clarity (Line 100-103) :

“FTIR and XRD analyzes have been used to identify the functional groups of the synthesized activated carbon surface, and the SEM images were applied to measure the pore structure of the synthesized carbon active”

Comment 6: In section 2.2.2, author mentioned on chemical activating agent of phosphoric acid and ZnCl2, but at the end didn’t conclude which one is being used in the methodology part. Amend the section accordingly.

Answer: 

Thank you very much for your comment. In this work ZnCl2 was applied as the activating agent. In order to clarify, the comment was applied and the manuscript was edited as follows (Line 153-157):

“To produce activated carbon, chemical activating agent of phosphoric acid and ZnCl2 can be used; in order to use a chemical process with phosphoric acid as a chemical activator, carbon material can be mixed with phosphoric acid, and then this mixture is put into a rotary kiln, after the carbonization, the activated carbon can be reached at ambient temperature. However, in this work ZnCl2 was applied as the activating agent.”

Comment 7: The section related to synthesis need to be revised with the addition of comparison of previous work in table form.

Answer: 

Thank you very much for your comment. The comment was applied and a table was added to the manuscript to compare the previous works results with this work (Table 2). The most important functional parameters of the synthesized adsorbents of the previous works, i.e., the specific surface area and the iodine number, are compared in Table 2.

Table (2) The comparison of Specific Surface and Iodine Number of activated carbon of the previous works

Adsorbent Specific Surface (m2/gr) Activation Method Iodine Number Ref

Cellulose Wastes 1100 Chemical 1080 (43)

Almond 900 Chemical 850 (44)

Walnut 840 Chemical 900 (44)

natural zeolite 198 Chemical 190 (45)

Oil sludge 120 Chemical 401 (46)

Comment 8: Should provide surface morphology images (SEM) to strengthen the activated carbon properties rather than just put in table.

Answer: 

Thank you very much for your comment. In this manuscript, an attempt has been made to conduct a comprehensive study on the synthesis of active carbon; for this reason, many experiments and analyzes have been performed and many results have been obtained, which were presented in the most possible compact way. However, in this manuscript, there are 12 figures and 8 tables; therefore, in order to reduce the number of tables and figures, SEM images have not been presented and only the results of the SEM analysis have been presented in one table along with other activated carbon properties.

Comment 9: More references need especially comparing the result in discussion.

Answer: 

Thank you very much for your comment. The comment was applied and more references (REF 43-48) were used in the results and discussion section to strengthen the results.

 

Reviewer 2

Comment 1: BET data should be supplemented.

Answer: 

Thank you very much for your comment. The specific surface area as a characteristic of synthesized activated carbons was presented in Table 1 which is determined using BET analysis. The manuscript lacked information on the method used to investigate the specific surface area. To address this, it was added that the BET analysis was applied to determine the specific surface area of the synthesized carbon actives. (Line 101-102)

Comment 2: The authors have performed the measurement at different temperatures. The authors may need to perform the required thermodynamic analysis, since the title has been stated so.

Answer: 

Thank you very much for your comment. As you mentioned, the manuscript aims to investigate the thermodynamics of oily wastewater treatment using activated carbon. To achieve this goal, isotherm models such as Langmuir, Freundlich, and Temkin have been used. Several studies have been conducted to understand the adsorption behavior of activated carbon using these models. The thermodynamic analysis of carbon active using these isotherm models provides a comprehensive understanding of the adsorption mechanism and helps in optimizing the adsorbent amount in a process. In other words, when the adsorbent is placed in contact with an oily solution, the oil concentration on the adsorbent surface increases until reaching thermodynamic equilibrium and then stabilizes at an equilibrium state. To understand the adsorption mechanism thermodynamically and to optimize the required adsorbent amount in a process, the isotherm models was applied. The obtained equilibrium value, known as the adsorption isotherm, is the primary basis for the design of adsorption systems. Adsorption isotherms are a powerful tool for the thermodynamic analysis of activated carbon; because the adsorption isotherms can be used to determine the adsorption capacity of activated carbon and to analyze the thermodynamics of the adsorption process.

Comment 3: How does "no-in depth" research on the cause of the difference in surface adsorption characteristics help in this study?

Answer: 

Thank you very much for your comment. It is important to note that the lack of in-depth research on a topic does not necessarily mean that there has been no research carried out on an area of study. It may indicate the need for further research and exploration in this area. In other words, this sentence means that the results of the works of other researchers should be placed next to each other and by covering the research gaps, a new and original result should emerge in order to open new horizons of science. Exactly what this work has tried to do to understand the causes of the difference in surface adsorption characteristics of the obtained carbon active with different precursors and different activation processes. However, in order to avoid misunderstanding, it was changed as below:

“Despite the numerous articles on activated carbon with different precursors in the adsorption process, more extensive research is needed to understand the causes of differences in the surface adsorption characteristics of activated carbon with different precursors and different activation processes.”

Comment 4: Based on "3", will the analysis be different if the method of activation is changed?

Answer: 

Thank you very much for your comment. In the syntheses of all types of adsorbents, the main goal is to produce adsorbents with a larger specific surface area and, as a result, greater potential for adsorption which can be studied using BET analysis and the iodine number determination, respectively. The most important functional parameters of the adsorbents of the previous works, i.e., the specific surface area and the iodine number, are compared in Table 2.

Comment 5: Formatting issue in Table 2.

Answer: 

Thank you very much for your comment. The comment was applied and the Table 2 (which is changed to Table 3 in the new manuscript) was edited properly.

Comment 6: How do the performance of the studied adsorbents reported as compared to the literature studies? 

Answer: 

Thank you very much for your comment. The comment was applied and a table was added to the manuscript to compare the previous works results with this work (Table 2). The most important functional parameters of the adsorbents of the previous works, i.e., the specific surface area and the iodine number, are compared in Table 2.

---

## [Decision Letter · Decision Letter 1]

6 Jun 2023

PONE-D-23-04431R1Thermodynamic and Structural Investigation of Oily Wastewater Treatment Using Peach Kernel and Walnut Shell Based Activated CarbonPLOS ONE

Dear Dr. Al-Ansari,

Thank you for submitting your manuscript to PLOS ONE. After careful consideration, we feel that it has merit but does not fully meet PLOS ONE’s publication criteria as it currently stands. Therefore, we invite you to submit a revised version of the manuscript that addresses the points raised during the review process.

We look forward to receiving your revised manuscript.

Kind regards,

Nor Adilla Rashidi, Ph.D.

Academic Editor

PLOS ONE

Journal Requirements:

Reviewers' comments:

Reviewer's Responses to Questions

**Comments to the Author**

1. If the authors have adequately addressed your comments raised in a previous round of review and you feel that this manuscript is now acceptable for publication, you may indicate that here to bypass the “Comments to the Author” section, enter your conflict of interest statement in the “Confidential to Editor” section, and submit your "Accept" recommendation.

Reviewer #1: All comments have been addressed

Reviewer #2: All comments have been addressed

2. Is the manuscript technically sound, and do the data support the conclusions?

Reviewer #1: Partly

Reviewer #2: Partly

3. Has the statistical analysis been performed appropriately and rigorously? 

Reviewer #1: I Don't Know

Reviewer #2: Yes

4. Have the authors made all data underlying the findings in their manuscript fully available?

Reviewer #1: Yes

Reviewer #2: Yes

5. Is the manuscript presented in an intelligible fashion and written in standard English?

Reviewer #1: Yes

Reviewer #2: Yes

6. Review Comments to the Author

Reviewer #1: The authors have addressed some of the comments thoroughly. However, some critical thinking might be required for the manuscript to stand out and having better clarity. As for the grammatical errors mentioned previously, the Abstract still in need of further refinement before being published.

Reviewer #2: The experimental section (particularly on the activation procedure) is a little too wordy. Probably should be a little more specific by removing irrelevant information (to the experimental) in the introduction (or result and discussion) instead.

7. PLOS authors have the option to publish the peer review history of their article (what does this mean?). If published, this will include your full peer review and any attached files.

Reviewer #1: **Yes: **Intan Syafiqah Ismail

Reviewer #2: No

---

## [Author Response · Author response to Decision Letter 1]

6 Dec 2023

1. We notice that your manuscript file was uploaded on February 18, 2023. Please can you upload the latest version of your revised manuscript as the main article file, ensuring that does not contain any tracked changes or highlighting. This will be used in the production process if your manuscript is accepted. Please follow this link for more information: http://blogs.plos.org/everyone/2011/05/10/how-to-submit-your-revised-manuscript/

Reply: We uploaded the last version of manuscript without any tracked changes or highlighted text.

2. Thank you for updating your data availability statement. You note that your data are available within the Supporting Information files, but no such files have been included with your submission. At this time we ask that you please upload your minimal data set as a Supporting Information file, or to a public repository such as Figshare or Dryad. Please also ensure that when you upload your file you include separate captions for your supplementary files at the end of your manuscript.

As soon as you confirm the location of the data underlying your findings, we will be able to proceed with the review of your submission.

Reply: We understand the importance of data accessibility and transparency in research. We would like to assure you that the data supporting the findings of our study are presented through the figures and tables included in the manuscript. These figures and tables contain the necessary information to reproduce the study results. However, we acknowledge that providing the data in a more accessible format would be beneficial for readers. To address this concern, we propose the providing the data upon reader request option for data availability. In this way, we are willing to share the data with interested readers upon request. We will ensure that the data are made available in a timely manner and provide the necessary contact information for readers to request access. We added below section to the paper:

Data Availability Statement: The data presented in this study are available on request from the corresponding author.

3. We note that the grant information you provided in the 'Funding Information' and 'Financial Disclosure' sections do not match.

When you resubmit, please ensure that you provide the correct grant numbers for the awards you received for your study in the 'Funding Information' section.

Reply: The exact information of funding and acknowledgment are as follow:

Acknowledgments:

The authors acknowledge the anonymous reviewers for their valuable suggestions that helped improve the quality of the manuscript.

Funding:

This work was funded by the King Khalid University, Abha, Saudi Arabia. The appreciation to the Deanship of Scientific Research at King Khalid University for funding this work through Large Groups Project under grant number (R.G.P. 2/57/44).

4. Please include a figure label and title for Figure 1 to 12 in your main manuscript.

Reply: We added label and title for all figures.

5. Please note that funding information should not appear in the Acknowledgments section or other areas of your manuscript. We will only publish funding information present in the Funding Statement section of the online submission form. Please remove any funding-related text from the manuscript.

Reply: We revised these two sections as follow:

Acknowledgments:

The authors acknowledge the anonymous reviewers for their valuable suggestions that helped improve the quality of the manuscript.

Funding:

This work was funded by the King Khalid University, Abha, Saudi Arabia. The appreciation to the Deanship of Scientific Research at King Khalid University for funding this work through Large Groups Project under grant number (R.G.P. 2/57/44).

6. We note that several of your files are duplicated on your submission. Please remove any unnecessary or old files from your revision, and make sure that only those relevant to the current version of the manuscript are included.

Reply: We removed unnecessary or old files from our revision that were duplicated.

---

## [Decision Letter · Decision Letter 2]

27 Dec 2023

Thermodynamic and Structural Investigation of Oily Wastewater Treatment Using Peach Kernel and Walnut Shell-Based Activated Carbon

PONE-D-23-04431R2

Dear Dr. Al-Ansari,

We’re pleased to inform you that your manuscript has been judged scientifically suitable for publication and will be formally accepted for publication once it meets all outstanding technical requirements.

Kind regards,

Nor Adilla Rashidi, Ph.D.

Academic Editor

PLOS ONE

Additional Editor Comments (optional):

Reviewers' comments:

Reviewer's Responses to Questions

**Comments to the Author**

1. If the authors have adequately addressed your comments raised in a previous round of review and you feel that this manuscript is now acceptable for publication, you may indicate that here to bypass the “Comments to the Author” section, enter your conflict of interest statement in the “Confidential to Editor” section, and submit your "Accept" recommendation.

Reviewer #2: All comments have been addressed

2. Is the manuscript technically sound, and do the data support the conclusions?

Reviewer #2: Yes

3. Has the statistical analysis been performed appropriately and rigorously? 

Reviewer #2: Yes

4. Have the authors made all data underlying the findings in their manuscript fully available?

Reviewer #2: Yes

5. Is the manuscript presented in an intelligible fashion and written in standard English?

Reviewer #2: Yes

6. Review Comments to the Author

Reviewer #2: Manuscript can be accepted in present form. The authors have tried to address to the best of the capability.

7. PLOS authors have the option to publish the peer review history of their article (what does this mean?). If published, this will include your full peer review and any attached files.

Reviewer #2: No

---

## [Editor Report · Acceptance letter]

2 May 2024

PONE-D-23-04431R2 

PLOS ONE

Dear Dr. Al-Ansari, 

I'm pleased to inform you that your manuscript has been deemed suitable for publication in PLOS ONE. Congratulations! Your manuscript is now being handed over to our production team.

Kind regards, 

on behalf of

Dr. Nor Adilla Rashidi 

Academic Editor

PLOS ONE